# MultiIoT: Towards Large-scale Multisensory Learning for the Internet of Things

## Abstract

The Internet of Things (IoT), the network integrating billions of smart physical devices embedded with sensors, software, and communication technologies for the purpose of connecting and exchanging data with other devices and systems, is a critical and rapidly expanding component of our modern world. The IoT ecosystem provides a rich source of real-world modalities such as motion, thermal, geolocation, imaging, depth, sensors, video, and audio for prediction tasks involving the pose, gaze, activities, and gestures of humans as well as the touch, contact, pose, 3D of physical objects. Machine learning presents a rich opportunity to automatically process IoT data at scale, enabling efficient inference for impact in understanding human wellbeing, controlling physical devices, and interconnecting smart cities. To develop machine learning technologies for IoT, this paper proposes MultiIoT, the most expansive IoT benchmark to date, encompassing over 1.15 million samples from 12 modalities and 8 tasks. MultiIoT introduces unique challenges involving (1) learning from many sensory modalities, (2) fine-grained interactions across long temporal ranges, and (3) extreme heterogeneity due to unique structure and noise topologies in real-world sensors. We also release a set of strong modeling baselines, spanning modality and task-specific methods to multisensory and multitask models to encourage future research in multisensory representation learning for IoT.

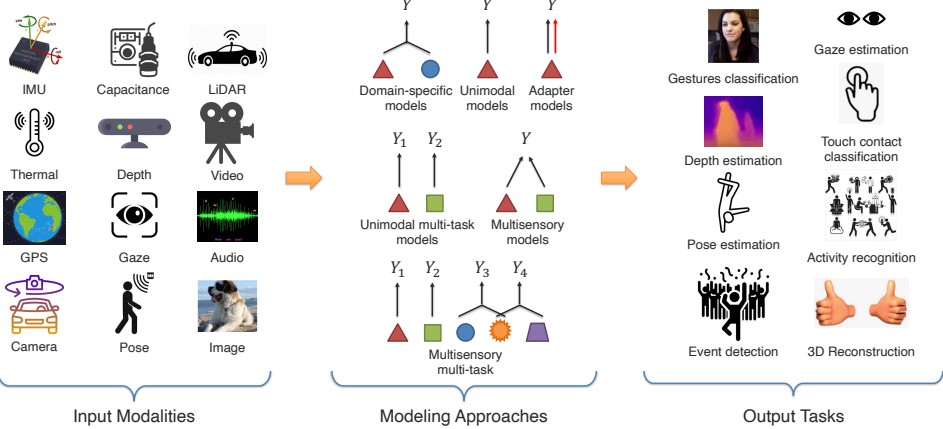

Figure 1: MultiIoT is a large-scale benchmark for representation learning on the Internet of Things (IoT), consisting of 1.15M samples, 12 rich modalities, and 8 challenging tasks in real-world physical scenarios. MultiIoT presents new challenges for impactful applications involving the pose, gaze, activities, and gestures of humans as well as the touch, contact, pose, 3D of physical objects.

## 1 Introduction

The digital world is witnessing an unprecedented surge in the realm of the Internet of Things (IoT): the ever-growing system of interlinked devices, communicating in tandem, from individual households to vast industrial complexes (Atzori et al., 2010). These devices are often embedded with sensors, software, and communication technologies that can safely and privately analyze the human and physical world (Rose et al., 2015; Li et al., 2015). For example, high-fidelity sensors are able to recognize physical activities to inform us of our daily physical wellness (Qi et al., 2015; Yuehong et al., 2016); vision, depth, and lidar sensors are able to navigate self-driving cards and connect them

with traffic lights for efficient transport management (Javaid et al., 2018; Khayyam et al., 2020); and wifi, depth, camera sensors able to detect if the elderly require assistance in hospitals (Ahamed & Farid, 2018; Kulkarni et al., 2014). The next generation of machine learning systems will need to understand and interact with the physical world through these IoT sensors. We call this new research field of discovering semantic structure and relationships in diverse physical sensors as *multisensory IoT*, since these sensors are inherently heterogeneous range and show interconnections. Enabling multisensory IoT at scale can drastically improve the accuracy and efficiency of processing IoT data and making autonomous predictions, which has great potential for impact in cities, hospitals, and the workplace (Liang et al., 2021a; Lee et al., 2022).

However, multisensory IoT data poses new challenges to our current methods for machine learning, since they introduce new heterogeneous real-world sensors with unique structure and noise topologies not typically seen in image-text research. Furthermore, real-world sensory data are often of long time scales, introducing challenges in long-range temporal interactions between modalities. Finally, multisensory IoT requires learning from many modalities and tasks to fully understand the physical world, which creates challenges in fusion and generalization.

In this paper, we make two contributions towards machine learning for IoT via:

- **MULTIIOT benchmark, modalities, and tasks**: We present the largest IoT representation learning dataset with over 1.15 million samples covering a rich set of 12 real-world sensory modalities. MULTIIOT define 8 IoT tasks, firmly rooted in practical scenarios such as personal wellness, healthcare, and smart cities, for broad impact in the physical world.
- **Empirical comparison of modeling paradigms**: To support machine learning research on MULTIIOT, we also release a set of strong modeling baselines, spanning modality and task-specific methods to multimodal and multitask models. Through extensive experiments, we highlight fundamental challenges in ML for IoT and summarize directions for future work in (1) learning from many sensory modalities, (2) long-range temporal interactions, and (3) extreme heterogeneity due to unique structure and noise topologies in real-world sensors.

## 2 Towards Multisensory Learning for IoT

The rapid expansion of the Internet-of-Things (IoT) landscape necessitates a comprehensive benchmark that captures the richness and variety of interconnected IoT devices to enable the development of robust representation learning methods. We introduce the MULTIIOT Benchmark - the largest and most diverse of its kind, comprising 1.15M samples spanning twelve distinct modalities and geared towards eight challenging tasks, as shown in Figure 1.

### 2.1 Technical challenges and selection criterion

We first formalize new challenges and potential real-world applications of representation learning for IoT that make it unique as compared to conventional representation learning. Based on these challenges, we describe our philosophy for the modalities and tasks selected in MULTIIOT, before providing extensive details on the final composed benchmark.

1. **High-modality multimodal learning**: While multimodal representation learning has historically been limited to image, text, video, and audio, real-world sensory modalities like IMU, thermal dynamics, GPS, depth, camera captures, audio, and more paint a more realistic picture of our multisensory physical world. These diverse modalities introduce new challenges in representation, fusion, and generalization across modalities, and create new opportunities for multitask and transfer learning across different physical sensors.
2. **Temporal interactions**: The second challenge lies in learning fine-grained multimodal interactions across long temporal ranges. Real-world sensory data is naturally sequential, possibly over extremely long time ranges, and multisensory sequential data may even show interactions between time steps that are not aligned. For example, typical image-text datasets have a sequence length of 77 words even lower (Lin et al., 2014), video datasets are roughly 10-60 seconds long, while MULTIIOT displays sequence lengths of up to 100-300 steps.
3. **Heterogeneity**: The third challenge is extreme heterogeneity in real-world sensors with unique structures and noise topologies. These sensory modalities may be naturally noisy or corrupted, not easily semantically segmented, and may not have natural language equivalents.

4. **Real-time**: Finally, many IoT devices need to run in real-time for applications in smart cities, security, healthcare, and automation. We therefore need to benchmark the efficiency of multisensory data collection, processing, and prediction as a critical quality alongside performance.

To reflect these 4 challenges in MULTIIoT, we amass data from various environments, IoT devices, and contexts, making it the most extensive available resource for IoT research.

## 2.2 TWELVE RICH MODALITIES

We collected diverse data from IoT devices, such as Inertial Measurement Units (IMU), Thermal sensors, Global Positioning Systems (GPS), capacitance, depth, gaze, and pose. We also collect commonly used image, audio, and video modalities in the physical world to bridge conventional multimodal research with the new challenges introduced by MULTIIoT.

**1) Inertial Measurement Units** capture 3D motion and orientation. This data is fundamental for various applications, including motion tracking and navigation. We include 2,940 IMU gaze samples (Kong et al., 2021) 28,400 IMU motion instances (Mollyn et al., 2022) 160,120 IMU samples (Arakawa et al., 2022), 330,178 IMU orientation recordings (Huang et al., 2018), and 510,142 timestamps-based IMU samples (Grauman et al., 2022).

**2) Thermal** modality data provide temperature radiance insights, crucial in surveillance. We used 12,025 Thermal samples from LLVIP (Jia et al., 2021) containing many pedestrians and cyclists from different locations on the street between 6 and 10 o'clock in the evening.

**3) Global Positioning Systems** offer location data with high precision. This data is invaluable for tasks like location-based services, asset tracking, and navigation. We include GPS data from self-driving cars using 41,000 samples from KITTI (Geiger et al., 2013) using OXTS RT3003 inertial and GPS navigation system for depth estimation. The geographic coordinates include global orientation, altitude, velocities, accelerations, angular rates, and satellite information.

**4) Cameras** capture the visual world in rich detail. We include 41,000 instances from KITTI self-driving car dataset (Geiger et al., 2013) using a Velodyne laser scanner installed on a vehicle car for depth estimation. The timestamp-based points can be considered according to the scanner's continuous rotation on its vertical axis, which provide context to GPS/IMU systems for auto-driving.

**5) Capacitance** sensors measure changes in capacitance to detect nearby objects or changes and are critical components of touchscreen technologies and proximity sensing. We used 65,374 samples from TouchPose (Ahuja et al., 2021) using a 39.6 cm capacitive Crystal Touch panel (Ocular Touch, Dallas, TX), 16-bit touch digitizer, and cameras to record ground-truth data. When fingers approach the lines on the mutual-capacitance touch sensor, it causes a capacitance drop between lines, resulting in the mutual-capacitance image.

**6) Depth** sensors measure distances between the sensor and objects, providing a 3D view of the environment. They play a significant role in tasks like object detection and scene reconstruction. We used 160,120 samples from RGBGaze (Arakawa et al., 2022) using Apple iPhone X with a TrueDepth camera ($640 \times 480$ depth map interpolated from a $170 \times 170$ IR dot pattern).

**7) Gaze** sensors track eye movement and direction, offering insights into user attention and intention. We used 2,940 samples from EyeMU (Kong et al., 2021) by running an iOS application on an Apple iPhone 12 Pro (screen size is $12.8 \times 6.4$ cm). The participants were asked to gaze at a single red dot, and the screen advanced to capture a motion gesture and a 2-axis gaze location after 1.2 seconds.

**8) Pose** sensors capture the orientation and position of objects or individuals critical for motion analysis and interactive applications. We include 330,178 samples from DIP-IMU (Huang et al., 2018) using Xsens IMU sensors. For touch pose data, we used 65,374 samples in TouchPose (Ahuja et al., 2021) from a Leap Motion stereo IR camera, running Orion 4.1 for 3D hand pose tracking. we included 14 different finger and whole-hand touch poses and gestures.

**9) LiDAR** sensors emit light to measure distances, generating high-resolution 3D maps of environments. They are central to autonomous driving and topographical mapping. We include 51,000 samples from the Newer College dataset (Ramezani et al., 2020) using the Ouster LiDAR with 64 beams, 64 Channels, 120 m range, $45°$ vertical Field-of-View (FoV), and 1024 horizontal resolution.

**10) Video** captures sequences of visual frames, providing a dynamic view of the environment. We used 510,142 egocentric videos in Ego4D (Grauman et al., 2022), which include many everyday

activities, such as cooking, cleaning, and fishing from diverse geographic locations across the world, and are paired with timestamps-based IMU values of the normalized accelerometer and gyroscopes.

**11) Audio** sensors capture sound waves, enabling voice recognition, sound classification, and environmental sound analysis. We include 28,400 samples from SAMoSA (Mollyn et al., 2022), where participants wore the smartwatch on their dominant arm, and were asked to perform 26 activities across 4 contexts with each activity repeated 3 times within each context.

**12) Image** sensors offer static visual captures of the environment, serving as a basis for a myriad of vision tasks. For RGB image data, we collected 160,120 samples from RGBDGaze (Arakawa et al., 2022) paired with gaze, depth, and IMU for gaze tracking, 41,000 samples from KITTI (Geiger et al., 2013), 12,025 high-quality images paired with infrared thermal samples in LLVIP (Jia et al., 2021), and 65,374 instances from TouchPose (Ahuja et al., 2021).

## 2.3 Eight Well-defined and Challenging Tasks

Upon these 12 sensory modalities, our benchmark also includes tasks that reflect real-world IoT challenges in order to drive the community towards solutions with tangible societal impacts.

**1) Gaze estimation:** This task is pivotal for human-computer interaction, driver monitoring, and virtual reality. Given RGB images of faces, depth and IMUs, our goal is to predict the location (X/Y) for tracking gazes of the person. This regression task requires multisensory understanding of long-range interactions between RGB images and depth and heterogeneity in IMUs.

**2) Depth estimation** involves predicting the distance between the camera and each pixel in the image and is a cornerstone for AR/VR applications, robotics, and object detection. Given RGB images, camera parameters, GPS coordinates, and IMU, we predict the depth maps of objects, such as cars and pedestrians on the streets. For robots, given RGB images, capacitive images, and hand poses, our target is to estimate the depth maps of left and right hands.

**3) Gesture classification:** Crucial for human-machine interfaces, gesture classification aims to recognize specific human hand or body movements. Given gaze locations and IMU data on accelerometer, gyroscope, and orientation, the goal is to classify human gestures. This classification problem requires the cross-modal perception of both gaze and IMUs.

**4) Pose estimation** focuses on determining the spatial arrangement of human joints and has applications in AR/VR, gaming, and health. Given RGB images and measured IMU data, our goal is to predict the poses of human body including 24 joints with three joint angles (yaw, pitch, roll). This regression problem requires fusing IMUs and RGB pixels.

**5) Touch contact classification** involves determining the type or nature of touch on capacitive surfaces, a vital component for enhancing user experiences on touch-based devices. Given RGB images, capacitive images, depth maps, and hand poses, the goal is to classify touch contact.

**6) Event detection:** A broad area with applications in health, wellness, smart homes, and the workplace, event detection involves identifying specific occurrences or anomalies in the data stream. Given audio spectrograms and IMU data of accelerometer, gyroscope, and orientation, our goal is to predict the categories of events across different timestamps. This classification problem requires modeling interactions between audio and IMU.

**7) Activity recognition:** Central to fitness, health, and elderly care, activity recognition aims to discern human activities like walking, running, or jumping. Given RGB images, poses with three joint angles (yaw, pitch, roll), and IMU data, we classify the class of actions for the human body. For egocentric cases, we are given video frames and IMU orientation recordings from different sensors to predict the category of activity in the videos. This classification task requires a cross-modal understanding of poses, videos, and IMU.

**8) 3D reconstruction** involves creating a three-dimensional model of an environment or object from 2D data, an application of huge significance in gaming, film, and AR/VR. Given RGB images, capacitance images, and depth maps, our target is to reconstruct the 3D poses. This regression problem requires a multimodal understanding of both capacitance images and depth maps.

## 3 Modeling Approaches

Given the extensive legacy of research in both IoT and ML, we aim to establish clear baselines that emerge from the confluence of both fields. We therefore include a wide spectrum of starting baseline models, across various tradeoffs in expressiveness, complexity, efficiency, and interpretability.

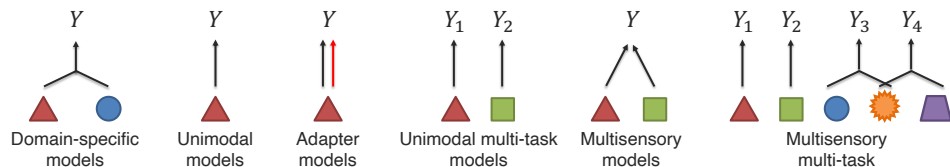

Figure 2: MULTIIOT includes a suite of benchmark models spanning (1) domain-specific models from IoT expert knowledge, (2) unimodal models most suitable for each modality, (3) adapter modules on top of pre-trained foundation models, (4) multitask models on the same modalities but for multiple tasks, (5) multisensory models for single tasks, and (6) multisensory multitask models that share information across many modalities and tasks.

**Domain-specific models.** Over the years, each sensor modality has evolved its own set of algorithms. For instance, IMU data has been traditionally processed using Kalman filters (Lai et al., 2019) to predict movement, while thermal modality often relies on image processing techniques for hotspot detection (George & Thampi, 2018; Zhu et al., 2021; Chockalingam et al., 2023). Before delving into advanced ML algorithms, it's crucial to acknowledge and learn from these traditional techniques that have been optimized for specific sensor outputs. A majority of IoT sensor data is inherently time-series (Atmoko et al., 2017; Khan et al., 2018; Kumar et al., 2020; Cook et al., 2020). Classical statistical methods like AutoRegressive Integrated Moving Average (ARIMA) (Lopez-Martin et al., 2020; Yasnita et al., 2020) or Exponential Smoothing have been employed to forecast, denoise, or detect anomalies in sensor readings (Gardner Jr., 1985; Billah et al., 2006; Alysha M. De Livera & Snyder, 2011). Their non-ML nature doesn't undermine their significance; instead, they serve as a solid foundation upon which more complex ML models can be built. IoT research has produced an array of techniques, ranging from signal processing methods, such as Fourier (Zhang et al., 2017; Murugan et al., 2017) and Wavelet Transforms (Muthukrishnan et al., 2019), to data compression (Hossain et al., 2019) and feature extraction strategies specific to resource-constrained devices (Ebrahimi et al., 2019; Khor et al., 2021; Imteaj et al., 2022). Many of these methods were designed to function efficiently in real-time scenarios with limited computational resources.

**Unimodal models** utilize a single data modality for a given task, ensuring that the model is strictly trained using information from one type of input (Kong et al., 2021; Mollyn et al., 2022). Each modality comes with its own unique characteristics. For instance, RGB images can capture the richness of color and texture, whereas depth maps provide information about distances and shapes. These unique types are leveraged to maximize the performance of the task. Given a dataset $D = \{(x_i, y_i)\}$, where $x_i$ is the input from a single modality and $y_i$ is the corresponding label, a unimodal model $M$ is trained to minimize the loss $L$:

$$L(D, M) = \sum_i \mathcal{L}(M(x_i), y_i). \tag{1}$$

**Adapter models** are modules inserted between the original layers of pre-trained models (Gao et al., 2023) to allow for fine-tuning specific to a new task, while maintaining the foundational knowledge of the pre-trained model. Given a pre-trained model with a set of weights $W$, and an adapter module $A$ with its own set of weights $W_A$, the output $y$ for an input $x$ is:

$$y = M_{W+A}(x) = M_W(A_{W_A}(x)). \tag{2}$$

where $M_{W+A}(\cdot), M_W(\cdot)$ denotes the model with both weights $W, A$ and weights $W$.

**Unimodal multi-task models** process the input data in a single encoder, and multiple decoders (one for each task) generate the outputs. The encoder is shared across tasks, aiming to capture the common features necessary for all tasks, while each decoder is task-specific. Given a dataset $D = \{(x_i, y_{i1}, y_{i2}, ...y_{in})\}$ where each $x_i$ has multiple corresponding labels for different tasks, the model minimizes a combined loss $L$:

$$L(D, M) = \sum_i \sum_j \mathcal{L}_j(M_j(E(x_i)), y_{ij}). \tag{3}$$

where $M_j(\cdot)$ denotes the $j$th task model, and $E$ denotes the encoders.

**Multisensory models** combine different modalities at some stage in the model – be it early fusion, middle fusion, or late fusion. A common approach is to use separate encoders for each modality and a shared decoder that fuses the representations to produce an output. Given multi-modal data

Table 1: Multisensory multi-task learning is a particularly effective approach on MULTIIoT, enabling information sharing to learn general representations for IoT data.

| Method | Gaze est. (cm, ↓) | Depth est. (mm, ↓) | Gesture cls. (%,↑) | Pose est. (cm, ↓) | Touch cls. (%, ↑) | Event det. (%, ↑) | Activity recog. (%, ↑) | 3D recons. (mm, ↓) |
|---|---|---|---|---|---|---|---|---|
| Domain-specific | 3.76 | 40.9 | 68.2 | 10.86 | 52.6 | 59.3 | 48.5 | 35.6 |
| Unimodal | 2.26 | 20.7 | 97.3 | 6.49 | 88.0 | 86.9 | 79.2 | 22.2 |
| Adapter | 2.05 | 18.6 | 97.6 | 5.75 | 88.7 | 87.5 | 82.3 | 21.3 |
| Unimodal Multi-task | 1.95 | 18.2 | 98.2 | 5.36 | 89.3 | 88.1 | 82.5 | 20.5 |
| Multisensory | 1.79 | 17.3 | 98.7 | 4.62 | 91.2 | 89.1 | 83.5 | 19.6 |
| Multisensory Multi-task | **1.08** | **13.6** | **99.3** | **3.85** | **93.8** | **92.7** | **87.5** | **17.5** |

$x = (x_1, x_2, ...x_m)$, the model combines representations:

$$y = T(E_1(x_1) \oplus E_2(x_2) \oplus ... \oplus E_m(x_m)) \tag{4}$$

where $T(\cdot)$ denotes the task head, and $E_1, E_2, ..., E_m$ denote the encoders.

**Multisensory multitask models** leverage data from different modalities and aims to solve more than one task simultaneously. It often benefits from potential correlations between tasks. For example, in an IoT setting, a model could use audio and visual data to simultaneously predict both the type of event occurring and its intensity. For multi-modal data $x = (x_1, x_2, ...x_m)$ and multiple tasks, the combined representations are processed as:

$$y_j = T_j(E_1(x_1) \oplus E_2(x_2) \oplus ... \oplus E_m(x_m)) \tag{5}$$

where $T_j(\cdot)$ denotes the $j$th task head, and $E_1, E_2, ..., E_m$ denote the encoders.

## 4 EXPERIMENTS

Our experiments aim to benchmark existing ML paradigms on MULTIIoT, including the best task-specific models as well as those designed for multimodal, multitask, long-range, and noisy data settings. We elaborate on the experimental setup and report our findings.

### 4.1 EXPERIMENTAL SETUP

All experiments were conducted on NVIDIA V100 GPUs. For **unimodal models**, data from each modality was processed independently using optimized neural architectures like CNNs for images. Models were trained with a batch size of 128, using the Adam optimizer at a learning rate of 0.001. **Adapter models** utilized deep architectures such as LLaMA-adpater with adapter layers, fine-tuning only the adapter modules with a learning rate of 0.0005. **Unimodal multi-task models** employed shared encoder layers and task-specific decoders, ensuring balanced gradients among tasks. For **multisensory models**, data fusion occurred at varying levels, from input to decision, and models ensured balanced data representation from each modality. Lastly, **multisensory multitask models** utilized modality-specific encoders followed by task-specific decoders, balancing both modalities and tasks during training. Each method's efficacy was validated on respective datasets.

To evaluate performance, we employ task-specific metrics. For gaze and pose estimation, we measure the mean euclidean error in centimeters between predictions and ground truth. Depth estimation utilizes mean absolute error in millimeters, while gesture classification, touch contact classification, and activity recognition rely on accuracy metrics. Event detection employs the F1 score for confident threshold predictions, and 3D pose reconstruction is assessed using the End-point-error in millimeters for joint discrepancies.

### 4.2 MAIN QUANTITATIVE RESULTS

**Overall performance**: Table 1 reports the quantitative results on MULTIIoT Benchmark using single modality, single task, multimodal multitask, and adapter and alignment models. As seen in Table 1, the multimodal multitask method consistently outperforms the single modality and single task models across all tasks. This can be attributed to their ability to integrate information across modalities and tasks, which is especially crucial when one modality might have noisy or incomplete data. The adapter and alignment models, while showing commendable performance due to their ability to adapt to new tasks, often fall short in scenarios where multiple modalities have to be processed simultaneously. The multimodal multitask method manages to strike a balance by leveraging the power of both multimodal inputs and multi-task training.

Table 2: Multimodal learning enables complementary learning of information and achieves strong performance on the MULTIIOT benchmark.

| Modality Ratio | Gaze est. (cm, ↓) | Depth est. (mm, ↓) | Gesture cls. (%,↑) | Pose est. (cm, ↓) | Touch cls. (%, ↑) | Event det. (%, ↑) | Activity recog. (%, ↑) | 3D recons. (mm, ↓) |
|---|---|---|---|---|---|---|---|---|
| single-modality | 2.26 | 20.7 | 97.3 | 6.49 | 88.0 | 86.9 | 79.2 | 22.2 |
| 25% | 2.13 | 19.6 | 97.5 | 5.97 | 88.9 | 87.3 | 80.2 | 21.5 |
| 50% | 1.95 | 18.7 | 98.1 | 5.38 | 90.1 | 88.2 | 81.3 | 20.9 |
| all | **1.79** | **17.3** | **98.7** | **4.62** | **91.2** | **89.1** | **83.5** | **19.6** |

Table 3: Multi-task learning is another effective strategy on the MULTIIOT benchmark, enabling information sharing across tasks.

| Task Ratio | Gaze est. (cm, ↓) | Depth est. (mm, ↓) | Gesture cls. (%,↑) | Pose est. (cm, ↓) | Touch cls. (%, ↑) | Event det. (%, ↑) | Activity recog. (%, ↑) | 3D recons. (mm, ↓) |
|---|---|---|---|---|---|---|---|---|
| single-task | 2.26 | 20.7 | 97.3 | 6.49 | 88.0 | 86.9 | 79.2 | 22.2 |
| 25% | 2.17 | 19.9 | 97.5 | 6.23 | 88.3 | 87.1 | 80.1 | 21.8 |
| 50% | 2.09 | 19.0 | 97.8 | 5.86 | 88.9 | 87.5 | 81.2 | 21.2 |
| all | **1.95** | **18.2** | **98.2** | **5.36** | **89.3** | **88.1** | **82.5** | **20.5** |

**Performance across different modalities**: In this section, we delve into experiments that focus solely on the efficacy of using multiple modalities as input, while keeping the task constant. Table 2 showcases the significant performance improvements observed when adopting a multimodal approach as opposed to unimodal setups by using various ratios (25%, 50%, all) of total modalities. For most tasks, the incorporation of multiple modalities resulted in more robust and accurate models. This can be attributed to the model's ability to tap into complementary information present in different modalities, especially in scenarios where one modality might be ambiguous or noisy.

**Performance across different tasks**: We separately analyze model performance when trained on multiple tasks simultaneously, while keeping the same modality inputs constant. Table 3 reveals that for most tasks, our multitask model's performance was on par with or exceeded that of models trained solely on individual tasks. This suggests that the shared representations learned during multitask learning were largely beneficial, since the model learns more generalized and robust features, while also improving computational efficiency.

**Zero-shot and few-shot transfer**: Finally, we study whether models trained on certain modalities or tasks can transfer to a new set of target modalities or tasks they have never seen during training (zero-shot) or have seen with only very few examples (few-shot). We chose the fix-8 dataset as the target, primarily because of its diverse representation of modalities (IMU, capacitance, depth, image) and its challenging task (gaze estimation and touch contact classification). We examined various configurations ranging from transferring an unimodal model to transferring the multimodal multitask models in Table 4. Few-shot performance was gauged by training the model on a minimal set of examples (5, 10, or 20) from the target task. Across the board, even a few examples significantly boosted performance compared to the zero-shot setting, which highlights the model's ability to quickly adapt to new information. The multimodal multitask models consistently outperformed other configurations. These gains were most pronounced in the 20-shot setting but were noticeably beneficial even in the 5-shot scenario. Our results suggest that multimodal and multitask training enables zero-shot and few-shot capabilities not seen in single-modality and single-task training, and is a promising approach to deal with limited labeled data often seen in real-world IoT systems.

## 4.3 UNDERSTANDING CHALLENGES IN MULTIIOT

**Testing long-range interactions**: Long-range interactions are critical to many problems in machine learning, particularly in fields like time series forecasting, natural language processing, and signal analysis. In a controlled experiment, we truncated sequences to various lengths and observed how conventional models performed. From Figure 3 (left), as the sequence lengths increased, representing longer durations of time

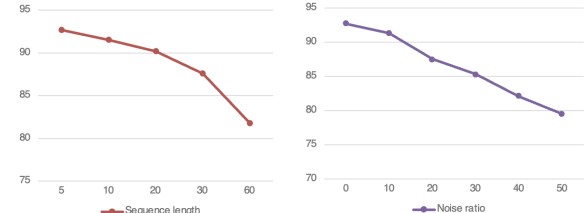

Figure 3: Long-range multimodal interactions and heterogeneity between modalities due to noise and imperfections make the MULTIIOT benchmark particularly challenging for ML models.

Table 4: Multimodal and multitask training enables zero-shot and few-shot capabilities, which is a promising approach to deal with limited labeled data often seen in real-world IoT systems.

| Method | Gaze estimation (cm, ↓) | Touch contact classification (%, ↑) |
|---|---|---|
| IMU | 2.65 | – |
| capacitance | – | 83.5 |
| depth | 2.45 | 86.2 |
| image | 2.26 | 88.0 |
| multimodal | 1.79 | 91.2 |
| multimodal multitask | **1.08** | **93.8** |
| multimodal multitask (zero-shot) | 2.18 | 88.6 |
| multimodal multitask (5-shot) | 1.96 | 89.5 |
| multimodal multitask (10-shot) | 1.89 | 90.2 |
| multimodal multitask (20-shot) | **1.81** | **91.1** |

or more extensive contexts, there was a marked decline in performance. This showcased the models' inability to effectively encapsulate and understand interactions beyond a certain range. Multimodal setups further complicate this when the long-range dependencies aren't just within a modality but can also be across modalities. Therefore, architectures that can handle both long-range and multisensory data will be critical for progress.

**Testing heterogeneity in structure and noise**: Differences in data distributions, both in terms of their natural structure and noise topologies, are a challenge for machine learning models. We evaluated the same models on datasets that combined structured data (such as GPS, IMU) with unstructured data (such as images or raw audio), and found that unimodal baselines often struggled to reconcile these different data forms, leading to a significant drop in accuracy. We also introduced varying degrees of noise into datasets, Gaussian noise in numerical data. From Figure 3 (right), all methods saw a rapid decline in performance as the noise levels increased. Future work should study models that can better handle heterogeneity in structure and noise.

### 4.4 Analysis of information sharing

Finally, we show examples of how information is shared across modalities and tasks, based on two potential sources of sharing: low-level modality features and high-level semantic concepts.

**Low-level modality features**: Different sensory modalities often contain unique low-level perceptual features that complement those in other modalities. We illustrate this information sharing across 3 modalities: IMU, video, and pose data for predicting 2 common activities: walking and dancing.

*Walking* is a common activity with distinctive rhythmic characteristics. Using IMU features, the model learns that rhythmic patterns, particularly in acceleration and deceleration, correspond to each walking step. The cadence, stability, and any irregularities in the walking pattern can also be inferred. Video features capture the holistic visual representation of walking, presenting details such as gait, arm swing, speed, stride length, and frequency. Finally, pose features highlight the specific posture changes during walking, emphasizing leg movement, foot placement, and body alignment.

*Dancing* requires complex and expressive motions with varying styles and dynamics. IMU provides dynamic, often non-linear patterns in IMU data, reflecting the dance's tempo, vigor, and style variations; video captures the dance form, style, synchronization, and expressiveness; and pose data captures the alignment and configuration of body parts, offering insights into dance postures, transitions, and intricate footwork or hand movements.

**High-level semantic concepts** encapsulate a more general conceptual understanding and reasoning about the environment. We show two examples showing how the audio and IMU modalities share information about two high-level semantic concepts, focusing on 'body pose' and 'hand pose'.

*Body pose* represents the spatial arrangement and posture of the entire human body. This can involve stances like standing, sitting, lying down, or even dynamic movements like jumping or running. For Audio, indirect cues such as the sound of footsteps, a person sitting down on a chair, or even the echo in a room (indicating a certain body pose affecting sound propagation) can provide hints about the body's posture. For IMU, accelerometers capture the directional movement while gyroscopes provide rotational dynamics to distinguish if a person is upright, moving rapidly, or stationary.

*Hand pose* looks at the orientation, gesture, and spatial arrangement of just the hands, ranging from gestures like waving, gripping, to more intricate signs in sign language. In audio, sounds like

clapping, snapping, or even the subtle rustling of hands moving through the air can be detected. The distinct sounds made by hang interactions with objects can also hint at specific hand poses. When IMU sensors are placed on the wrist or back of the hand, they can capture detailed dynamics of hand movements, tilting, rotation, or swift movements that indicate hand poses.

## 5 RELATED WORK

**Internet of Things (IoT) representation learning** aims to discover semantic structure and relationships from vast streams of heterogeneous and multisensory physical data. For example, DIP-IMU (Huang et al., 2018) fuses depth sensing and IMU data for pose estimation. EyeMU (Kong et al., 2021) harnesses IMU sensors data using time-series neural networks for gaze tracking on mobile devices. TouchPose (Ahuja et al., 2021) designs approaches for tactile data, LLVIP (Jia et al., 2021) uses visual IoT data in dynamic real-world environments, and RGBDGaze (Arakawa et al., 2022) use RGBD data for gaze estimation. While these methods have creatively applied ML models, they are limited to individual tasks and are mostly unimodal in nature. Bringing a wide spectrum of related IoT modalities and tasks together enables statistical sharing using general models.

**Multimodal transformers** are strong models for temporal and multisensory data with their ability to automatically align and capture interacting features at different time-steps (Tsai et al., 2019; Yao & Wan, 2020; Lee et al., 2020). Self-supervised multimodal pretraining has emerged as an effective way to train these architectures, where representations are learned from large-scale unlabeled multimodal data before transferring to downstream tasks via fine-tuning (Lu et al., 2019; Li et al., 2019; Su et al., 2020). These pretraining objectives typically consist of unimodal masked prediction, crossmodal masked prediction, and multimodal alignment prediction (Hendricks et al., 2021).

**Multimodal multitask learning** aims to learn general-purpose representations for many modalities and tasks. Several works such as Perceiver (Jaegle et al., 2021a;b), MultiModel (Kaiser et al., 2017), ViT-BERT (Li et al., 2021), and PolyViT (Likhosherstov et al., 2022) have explored using the same architecture for different inputs on unimodal tasks (i.e., language, image, video, or audio-only). HighMMT (Liang et al., 2022) and Gato (Reed et al., 2022) learn higher-order interactions for multiple modalities and tasks to achieve strong results across multiple tasks. This line of research has also been enabled by high-quality benchmarks with diverse modalities, such as MultiBench (Liang et al., 2021b), MME (Fu et al., 2023), MultiInstruct (Xu et al., 2022), and OpenFlamingo (Awadalla et al., 2023). Our goal is to do the same for multisensory data in the physical world.

## 6 CONCLUSION AND BROADER IMPACTS

This paper proposes MULTIIOT, the most expansive IoT benchmark to date, encompassing over 1.15 million samples from 12 modalities and 8 tasks. MULTIIOT introduces unique challenges involving (1) learning from many sensory modalities, (2) fine-grained multisensory interactions across long temporal ranges, and (3) extreme heterogeneity due to ambiguous semantic abstraction and unique noise topologies in real-world sensors, which inspire several directions for future not encountered in conventional representation learning research. MULTIIOT, our standardized code, and leaderboards are publicly available, will be regularly updated, and welcome inputs from the community. We are aware of some potential **limitations and broader impacts**:

1. **Data privacy**: There may be privacy risks associated with making predictions from multimodal data of recorded human behaviors, such as video, audio, activities, poses, and wearable sensors. Datasets are collected from participants who have consented to data release. We only use these datasets for research purposes. All data was anonymized and stripped of all personal (e.g., personally identifiable information) and protected attributes (e.g., race, gender).
2. **Real-world privacy**: To deploy these algorithms at scale in the real world, it is also important to keep data and features private on each device without sending it to other locations using techniques such as federated learning (Li et al., 2018; Liang et al., 2020), differential privacy (Geyer et al., 2017), or encryption (Dankar & El Emam, 2013).
3. **Efficiency**: Modern ML models can cause environmental impacts resulting from the carbon footprint required to run large-scale models. ML for IoT can inspire the design of lightweight models that can run efficiently on edge devices and low-cost sensors (Strubell et al., 2019).
4. **Biases**: We also acknowledge that there is a risk of exposure bias due to imbalanced datasets, especially when human-centric data and possibly sensitive labels are involved. Models trained on biased data have been shown to amplify the underlying social biases (Lloyd, 2018). It is crucial for future work to mitigate social biases for sensory modalities and multimodal models.

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

APPENDIX

## A  DETAILED BENCHMARK

We introduce the MULTIIOT Benchmark - the largest and most diverse of its kind, comprising 1.15M samples spanning twelve distinct modalities and geared towards eight challenging tasks, as shown in Figure 1.

### A.1  TECHNICAL CHALLENGES AND SELECTION CRITERION

We first formalize new challenges and potential real-world applications of representation learning for IoT that make it unique as compared to conventional representation learning. Based on these challenges, we describe our philosophy for the modalities and tasks selected in MULTIIOT, before providing extensive details on the final composed benchmark.

1. **High-modality**: The first challenge is in high-modality multimodal learning from diverse modalities and tasks. While multimodal representation learning has historically been limited to image, text, video, and audio, MULTIIOT is designed to include IMU sensors (Huang et al., 2018), thermal dynamics (Jia et al., 2021), GPS signals, camera captures, and more to fully simulate a more realistic environment of our physical world where multiple data streams coexist. Different modalities can provide complementary information. For example, while audio can capture spoken words, visual data can capture facial expressions and gestures, enriching the overall context. These diverse modalities introduce new challenges in representation, fusion, and generalization across modalities, and create new opportunities for multitask and transfer learning across different physical sensors.

2. **Temporal**: The second challenge lies in learning fine-grained multimodal interactions across long temporal ranges. Real-world sensory data is naturally sequential, possibly over extremely long time ranges. Multisensory sequential data shows interactions that may not be aligned at the same time, which introduces a long-range temporal component as compared to conventional image-text research. For example, typical image-text datasets have a sequence length of 77 words even lower (Lin et al., 2014), video datasets are roughly 10-60 seconds long, while MULTIIOT displays sequence lengths of up to 100-300 steps.

3. **Real-world**: The final challenge is extreme heterogeneity in real-world sensors with ambiguous semantic abstraction and unique noise topologies. MULTIIOT contains data modalities that are naturally noisy, and may not have natural language equivalences, which prevents easy approaches in conditioning language models. Finally, we need to test the robustness of our models when different sensors might not be available or may be corrupted.

4. **Real-time**: IoT devices, given their proliferation, are becoming significant sources of data in smart cities, homes, and industries. Many IoT applications, from healthcare and entertainment to security and automation, require real-time multimodal data processing and real-time prediction. We therefore need to benchmark efficiency as a critical quality alongside performance measures.

To reflect these 4 challenges in MULTIIOT, we amass data from various environments, IoT devices, and contexts, making it the most extensive available resource for IoT research.

### A.2  TWELVE RICH MODALITIES

We collected diverse data from IoT devices, such as Inertial Measurement Units (IMU), Thermal sensors, and Global Positioning Systems (GPS). Furthermore, we include challenging modalities, such as capacitance, depth, gaze, and pose. Finally, we collect common and widely used image, audio, and video modalities. These modalities bring unique challenges since they typically involve noisy real-world sensor measurements, that lack explicit tokenization and alignment with other modalities that we typically expect from conventional multimodal image-text research.

**IMU:** Inertial Measurement Units capture 3D motion and orientation. This data is fundamental for various applications, including motion tracking and navigation. We collected 2,940 IMU samples from EyeMU (Kong et al., 2021) for gaze estimation and motion gesture classification, where they used the accelerometer and gyroscope raw values sampled at 60 Hz as the IMU values per-axis. 28,400 IMU instances are included from SAMoSA (Mollyn et al., 2022) to save synchronized streams of the 9-axis IMU data (accelerometer, gyroscope and orientation) at 50 Hz by using a Fossil Gen 5 smartwatch running Google Android wearOS 2.23. Further, we sampled 160,120 IMU sam-

ples (9-axis) recorded by the device motion sensor using an iOS application Arakawa et al. (2022) on Apple iPhone X for gaze tracking. For human bodies, 330,178 IMU orientation recordings (Huang et al., 2018) from 17 sensors on different body parts are saved for pose estimation and activity recognition. For first-person videos in Ego4D (Grauman et al., 2022), we used 510,142 timestamps-based IMU samples with the normalized accelerometer and gyroscope values in each video for activity recognition. For IMU data on self-driving cars, we collected 41,000 samples from KITTI (Geiger et al., 2013) for depth estimation.

**Thermal:** Thermal modality data provide temperature radiance insights, crucial in surveillance. For collection, we used 12,025 Thermal samples from LLVIP (Jia et al., 2021) containing many pedestrians and cyclists from different locations on the street between 6 and 10 o'clock in the evening. They used HIKVISION DS-2TD8166BJZFY-75H2F/V2 as the camera equipment, a binocular camera platform consisting of an infrared camera with a wavelength of $8 \sim 14$um.

**GPS:** Global Positioning Systems offer location data with high precision. This data is invaluable for tasks like location-based services, asset tracking, and navigation. For GPS data on self-driving cars, we collected 41,000 samples from KITTI (Geiger et al., 2013) using OXTS RT3003 inertial and GPS navigation system for depth estimation. The geographic coordinates include global orientation, altitude, velocities, accelerations, angular rates, and satellite information. Following the original dataset, we applied two specified 3-axis coordinates as accelerations for the vehicle and angular rates to describe the tangent plane of the earth's surface corresponding to the geographic location.

**Camera:** Cameras provide visual data, capturing the environment in rich detail. They serve as the backbone for countless computer vision tasks. For Camera data, we collected 41,000 instances from KITTI (Geiger et al., 2013) using a Velodyne laser scanner installed on a vehicle car for depth estimation. We stored its 3-axis coordinate and an additional reflectance value for each point. The timestamp-based points can be considered according to the scanner's continuous rotation on its vertical axis, which provides a complementary context to GPS/IMU systems for auto-driving.

**Capacitance:** Capacitive sensors measure changes in capacitance to detect nearby objects or changes. This is foundational for touchscreen technologies and proximity sensing. For capacitance data, we used 65,374 samples from TouchPose (Ahuja et al., 2021) using a 39.6 cm capacitive Crystal Touch panel (Ocular Touch, Dallas, TX), 16-bit touch digitizer, and cameras to record ground-truth data. When fingers approach the lines on the mutual-capacitance touch sensor, it causes a capacitance drop between lines, resulting in the mutual-capacitance image.

**Depth:** Depth sensors measure distances between the sensor and objects, providing a 3D view of the environment. They play a significant role in tasks like object detection and scene reconstruction. For depth data, we collected 160,120 samples from RGBGaze (Arakawa et al., 2022) using Apple iPhone X with a TrueDepth camera ($640 \times 480$ depth map interpolated from a $170 \times 170$ IR dot pattern). The participants were asked to look at a target (red dot) that was moving on the screen. While the user gazed at the target, the depth imagery was logged at approximately 8 Hz, along with the ARKit gaze prediction. For touch depth data, we used 65,374 samples from TouchPose (Ahuja et al., 2021) recorded by a ToF depth camera (Azure Kinect) below the surface of the transparent touch panel. The depth modality is essential for touch contact and 3D pose joint reconstruction.

**Gaze:** Gaze sensors track eye movement and direction, offering insights into user attention and intention. Regarding gaze modality, we collected 2,940 samples from EyeMU (Kong et al., 2021) by running an iOS application on an Apple iPhone 12 Pro (screen size is $12.8 \times 6.4$ cm). The participants were asked to gaze at a single red dot, and the screen advanced to capture a motion gesture and a 2-axis gaze location after 1.2 seconds. Furthermore, we used 160,120 gaze samples from RGBGaze (Arakawa et al., 2022) by running an iOS application on Apple iPhone X to record gaze tracking data with pre-determined 35 fixed locations on the screen.

**Pose:** Pose sensors capture the orientation and position of objects or individuals, which is critical for motion analysis and interactive applications. For body pose data, we collected 330,178 samples from DIP-IMU (Huang et al., 2018) using Xsens IMU sensors containing 3-axis accelerometers, gyroscopes, and magnetometers. They placed the head sensor onto the head of each participant such that the sensor axes aligned with the SMPL body frame to do a calibration. The SMPL pose parameters are stored in the angle-axis format with three joint angles (yaw, pitch, roll) per 24 joints. For touch pose data, we used 65,374 samples in TouchPose (Ahuja et al., 2021) from a Leap Motion

stereo IR camera, running Orion 4.1 for 3D hand pose tracking. We included 14 different finger and whole-hand touch poses and gestures, representing each pose with 3030 to 5875 samples.

**LiDAR:** LiDAR sensors emit light to measure distances, generating high-resolution 3D maps of environments. They are central to applications like autonomous driving and topographical mapping. For LiDAR data, we collected 51,000 samples from the Newer College dataset (Ramezani et al., 2020) using the Ouster LiDAR with 64 beams, 64 Channels, 120 m range, 45° vertical Field-of-View (FoV), and 1024 horizontal resolution. During the collection, the Ouster LiDAR synchronized with the recording computer using the Precision Time Protocol (PTP) to achieve sub-microsecond accuracy. The accurate prior map of 290 million points was down-sampled to 1 cm resolution and reduced to about 17 million points, allowing for its use without an observable drop in registration accuracy. Further, cropping the reduced point cloud around the sensor's pose dynamically created the final reference cloud in 100 m by 100 m.

**Video:** Video captures sequences of visual frames, providing a dynamic view of the environment. This modality supports tasks ranging from action recognition to anomaly detection. For video modality, we used 510,142 egocentric videos with 30FPS in Ego4D (Grauman et al., 2022), which includes a wide range of everyday activities, such as cooking, cleaning, and fishing. These videos also cover diverse geographic locations across the world, and are paired with timestamps-based IMU values of the normalized accelerometer and gyroscopes.

**Audio:** Audio sensors capture sound waves, enabling voice recognition, sound classification, and environmental sound analysis. For audio data, we collected 28,400 samples paired with IMU modality from SAMoSA (Mollyn et al., 2022), where participants wore the smartwatch on their dominant arm, and were asked to perform 26 activities across 4 contexts with each activity repeated 3 times within each context. As the audio was sampled at 1 kHz, the resolution of the information went down, and more activity classes, such as Hand Washing and Toothbrushing, got similar and confused. In such cases, IMU data can provide valuable information to remove ambiguity.

**Image:** Image sensors offer static visual captures of the environment, serving as a basis for a myriad of vision tasks. For RGB image data, we collected 160,120 samples from RGBDGaze (Arakawa et al., 2022) paired with gaze, depth, and IMU for gaze tracking. To align GPS and Camera modalities with images, we collected 41,000 samples from KITTI (Geiger et al., 2013) for depth estimation and activity recognition. Furthermore, we used 12,025 high-quality images paired with infrared thermal samples in LLVIP (Jia et al., 2021) from 26 locations. For alignment with body pose, we used 330,178 samples from DIP-IMU (Huang et al., 2018) for pose estimation and activity recognition. Regarding hand pose images, we collected 65,374 instances from TouchPose (Ahuja et al., 2021) for touch contact classification and 3D hand pose joint reconstruction.

### A.3 EIGHT WELL-DEFINED AND CHALLENGING TASKS

Our benchmark includes tasks that reflect real-world IoT challenges and that will drive the community towards solutions with tangible societal impacts.

**Gaze estimation:** This task is pivotal for human-computer interaction, driver monitoring, and virtual reality. Given RGB images of faces, depth and IMUs, our goal is to predict the location (X/Y) for tracking gazes of the person. This regression task requires multisensory understanding on long-range interactions between RGB images and depth and heterogeneity in IMUs.

**Depth estimation:** A cornerstone for AR/VR applications, robotics, and object detection, depth estimation involves predicting the distance between the camera and each pixel in the image. Given RGB images, camera parameters, GPS coordinates, and IMU, we are expected to predict the depth maps of objects, such as cars and pedestrian on the streets. In the touch robots case, given RGB images, capacitive image, and hand poses, our target is to estimate the depth maps of hands. This regression problem requires multisensory understanding on long-range interactions between RGB images and capacitance and heterogeneity in poses.

**Gesture classification:** Crucial for intuitive human-machine interfaces, gesture classification aims to recognize specific hand or body movements. Given gaze locations and IMU data on accelerometer, gyroscope and orientation, the task is defined to classify the gesture of human heads. This classification problem requires the cross-model perception on heterogeneity in gaze and IMUs.

**Pose estimation:** With applications in AR/VR, gaming, and health, pose estimation focuses on determining the spatial arrangement of human joints. Given RGB images and measured IMU data, our goal is to predict the poses of human body including 24 joints with three joint angles (yaw, pitch, roll). This regression problem requires a deeper cross-modal understanding on the heterogeneity in IMUs and RGB pixels.

**Touch contact classification:** Vital for enhancing user experiences on touch-based devices, this task involves determining the type or nature of touch on capacitive surfaces. Given RGB images, capacitive images, depth maps, and hand poses, we are expected to classify touch contact using diverse modalities. This classification task requires a multimodal understanding on the long-range interactions between RGB images and capacitance and heterogeneity in depth maps and poses.

**Event detection:** A broad area with applications in surveillance, smart homes, and industrial setups, event detection involves identifying specific occurrences or anomalies in the data stream. Given audio spectrograms and IMU data on accelerometer, gyroscope and orientation, our goal is to predict the categories of events across different timestamps. This classification problem requires a cross-modal understanding on the long-range interactions between audio and IMU. If a predicted activity is above a confidence threshold, we consider it an event. Othwise, if it's below a confidence threshold, or belongs to the Other class, we do not consider it an event.

**Activity recognition:** Central to fitness, health, and elder care applications, activity recognition aims to discern human activities like walking, running, or jumping. Given RGB images, poses with three joint angles (yaw, pitch, roll), and IMU data, we are expected to classify the class of actions for the human body. For ego-centric cases, we are given video frames and IMU orientation recordings on from different sensors to predict the category of activity in the videos. This classification task requires a cross-modal understanding on the heterogeneity in poses, videos and IMU.

**3D reconstruction:** With significance in gaming, film, and AR/VR, 3D reconstruction involves creating a three-dimensional model of an environment or object from 2D data. Given RGB images, capacitance image, and depth maps, our target is to reconstruct the 3D poses. This regression problem requires a multimodal understanding of both capacitance images and depth maps.

## B    EXPERIMENTAL SETUP

### B.1    SETUP FOR UNIMODAL MODELS

- **Data Preparation:** Each modality, e.g., RGB images, capacitive images, or hand pose, is pre-processed independently. The data undergo normalization and any specific transformations tailored to that modality.

- **Network Architecture:** Distinct neural architectures optimized for each modality type, such as CNNs for images and RNNs for sequential data.

- **Training Details:** Models are trained using a batch size of 128, employing the Adam optimizer with a learning rate of 0.001. Early stopping with a patience of 10 epochs ensures prevention from overfitting.

- **Evaluation:** Each unimodal model is evaluated on its respective validation dataset to gauge performance.

### B.2    SETUP FOR ADAPTER MODELS

- **Data Preparation:** Data is fed through a pre-trained network, where only the adapter modules are trainable.

- **Network Architecture:** Utilizing deep architectures like LLaMA (Gao et al., 2023), but with adapter layers inserted in-between the pre-defined layers.

- **Training Details:** Since only the adapter layers are trainable, fewer parameters are updated, allowing for a larger batch size of 256. The training uses the Adam optimizer with a learning rate of 0.0005.

- **Evaluation:** Model performance is assessed by evaluating the fine-tuned model on the targeted task's validation set.

### B.3 SETUP FOR UNIMODAL MULTI-TASK MODELS

- **Data Preparation:** Data from different tasks, but the same modality, are concatenated or paired.

- **Network Architecture:** Shared encoder layers process the input data, followed by task-specific decoders.

- **Training Details:** Gradient balancing techniques are employed to prevent one task from dominating the training process. Training leverages a batch size of 128 and the Adam optimizer with a learning rate of 0.001.

- **Evaluation:** Performance is evaluated separately for each task on their respective validation sets.

### B.4 SETUP FOR MULTISENSORY MODELS

- **Data Preparation:** Data from different modalities are fused either at the input, feature, or decision level.

- **Network Architecture:** Modality-specific encoders process each input type. Fusion layers then combine features from all encoders.

- **Training Details:** Models are trained with a batch size of 128 using the Adam optimizer and a learning rate of 0.001. Data balancing techniques ensure equal representation from each modality.

- **Evaluation:** The combined model's efficacy is evaluated using a validation dataset that includes all modalities.

### B.5 SETUP FOR MULTISENSORY MULTITASK MODELS

- **Data Preparation:** Data from different modalities and tasks are paired or concatenated as required.

- **Network Architecture:** Shared modality-specific encoders are followed by task-specific decoders.

- **Training Details:** Gradient balancing techniques are applied, along with modality balancing, to ensure fairness in learning. The model trains using a batch size of 128 and the Adam optimizer at a learning rate of 0.001.

- **Evaluation:** Each task's performance is assessed on their respective validation datasets.

For all the methods, the experimental environment remains consistent. All models are trained and evaluated on NVIDIA V100 GPUs, ensuring uniformity in computational power and performance.

## C EVALUATION METRICS

To measure performance, we utilize a combination of metrics following prior work on each specific task. For gaze estimation, we use mean euclidean error in centimeters to measure the positional distance between the predicted gaze and the ground-truth gaze. For depth estimation, we apply mean absolute error in millimeter to calculate the gap between the prediction and the ground-truth depth. For gesture classification, we compute the ratio of correct classified samples as the accuracy. For pose estimation, we use mean euclidean error in centimeters to measure the positional distance between the predicted pose joints and the ground-truth pose. For touch contact classification, we calculate the accuracy of classifying the category of fingers interacting with the touchscreen. For event detection, we apply F1 score to decide if the predicted activity above a confident threshold belongs to a event. For activity recognition, we compute the balanced accuracy for measuring instance-level performance. For 3D pose reconstruction, we use End-point-error in millimeter, the mean Euclidean error between all the joints of the annotated and predicted hand pose.

# D    MORE ANALYSIS

**Testing long-range interactions**: Long-range interactions are critical to many problems in machine learning, particularly in fields like time series forecasting, natural language processing, and signal analysis. Recognizing patterns and relationships over vast sequences or across multiple modalities often requires models to understand and leverage these long-range dependencies. However, capturing these interactions remains a challenge for many conventional models.

In a controlled experiment, we truncated sequences to various lengths and observed how conventional models performed. As the sequence lengths increased, representing longer durations of time or more extensive contexts, there was a marked decline in performance. This showcased the models' inability to effectively encapsulate and understand interactions beyond a certain range. Multimodal setups further complicate this. The long-range dependencies aren't just within a modality but can also be across modalities. This inter-modality long-range interaction is a largely uncharted territory, and our experiments showed that it's an area where even advanced models can falter.

Exploring architectures that inherently focus on long-range interactions, potentially leveraging self-attention mechanisms but with modifications to handle extremely long sequences. Employing models that operate at different temporal scales, allowing them to summarize information at various levels and potentially capture longer-range interactions more effectively. Techniques that allow models to allocate more computational resources when faced with potential long-range dependencies, thus emphasizing critical parts of a sequence or modality. For multimodal problems, mechanisms that facilitate better cross-modal attention can be crucial. This will enable models to recognize and act upon dependencies that span across different modalities, even if they are separated by considerable temporal or sequential gaps.

**Testing heterogeneity in structure and noise**: Heterogeneity in data, both in terms of structure and noise, is a pervasive challenge in machine learning. As datasets grow more complex, encompassing a wider variety of sources, the inherent differences in data structure and the presence of various types of noise can significantly hamper the performance of models. Understanding how models grapple with such heterogeneity is vital for real-world applications.

We exposed models to datasets that combined structured data (such as GPS, IMU) with unstructured data (such as images or raw audio). Unimodal baselines often struggled to reconcile these different data forms, leading to a significant drop in accuracy compared to when dealing with homogenous data types. We also introduced varying degrees of noise into datasets, Gaussian noise in numerical data. Currrent methods saw a rapid decline in performance as the noise levels increased, unable to filter out irrelevant information effectively. Heterogeneity challenges underline the importance of robustness in model design. Our experiments highlighted that many models, even those considered state-of-the-art, have vulnerabilities when exposed to unexpected data structures or noise patterns.

Exploring architectures and training techniques that are inherently more robust to noise and heterogeneity. This might include noise injection during training or techniques like dropout that encourage model generalization. Leveraging advanced data augmentation techniques, both for structured and unstructured data, to simulate and thus prepare the model for varied data structures and noise patterns. Using meta-learning approaches where models are trained to quickly adapt to new data structures or noise patterns with minimal fine-tuning. Research into advanced denoising mechanisms, especially ones that can handle structured noise, can be invaluable. This includes both pre-processing methods and in-model techniques.

