# OpenReview forum: "MultiIoT: Towards Large-scale Multisensory Learning for the Internet of Things"
_ICLR.cc/2024/Conference — Submitted to ICLR 2024_

### Official Review · Reviewer_4SPn · 2023-10-29

**Soundness:** 2 fair
**Presentation:** 2 fair
**Contribution:** 2 fair
**Rating:** 5
**Confidence:** 5

**Summary:**

This paper proposes MultiIOT which includes over 1.15 million samples from 12 modalities and 8 tasks. This paper summarizes the recent developments and key challenges in the field. Then, the authors benchmark the different model architectures for processing multi-modal sensory signals and propose some insights.

**Strengths:**

- I like the author's efforts in incorporating more modalities in understanding the scenes and human behaviors. This work is in general well-motivated and I believe this work would be interesting for future ML research from an application standpoint.
- Discussions on current situations in the field and outstanding challenges are well-written and easy to follow.

**Weaknesses:**

The main weaknesses of this work are the technical contribution and experimental evaluation.
- For dataset and benchmark, in Sec. 2.2 the authors claim 'We collected diverse data from IoT devices, such as Inertial Measurement Units (IMU), Thermal sensors, Global Positioning Systems (GPS), capacitance, depth, gaze, and pose.' However, from my understanding, it consists of solely existing datasets while most of them contain only several modalities.
- The experiments section contains no quantitative comparison with existing methods. There are other methods proposed for these individual tasks, and it would be difficult to evaluate the performance of the evaluated model variants without comparing them with the existing baselines.

**Questions:**

1. Can the authors discuss if there is extra effort in consolidating the different datasets? e.g. how to unify the data format and make them really 'one' benchmark and convenient for the research community to benchmark their algorithms on all the tasks easily.
2. What are the implementation details for each task? In Sec. B there is some brief explanation like 'Network Architecture: Distinct neural architectures optimized for each modality type, such as CNNs for images and RNNs for sequential data', but it is not enough. More experimental details are needed to understand and replicate the experiments.

Minor Issues:
- No qualitative results were provided. Authors could consider including data points and visualizations for the dataset, benchmark, and method.
- Fig. 3 is of low visual quality. Authors could design more and better charts to illustrate the model comparisons.

---

### Official Review · Reviewer_18YP · 2023-10-29

**Soundness:** 2 fair
**Presentation:** 3 good
**Contribution:** 2 fair
**Rating:** 3
**Confidence:** 4

**Summary:**

This paper provides an extensive benchmark, MultiIoT, for machine learning of IoT applications, that contains a large amount of data samples from 12 modalities and 8 different downstream tasks. Experiments are provided to compare the performance of machine learning models trained on different learning objectives and sensory modalities on each task. The conclusion was made that multi-modal and multi-task learning is beneficial in learning useful semantics from each modality.

**Strengths:**

1. The coverage of the sensory modalities and downstream tasks in the paper is fairly comprehensive.

2. Some of the observations made in the paper are interesting and can motivate future research in the IoT domain. For example, the authors found that the interaction of different tasks can facilitate single-task performance

**Weaknesses:**

1. As a benchmark, the authors did not provide a new dataset with comprehensive modality and task coverage that can be used for general IoT machine-learning models. Instead, the datasets evaluated in the benchmark all come from public resources, which only contain a subset of sensory modalities. For this reason, I feel it is actually an overclaim to address that the benchmark consists of over 1.15M samples, which comes from the sum of different datasets.

2. The paper lacks a thorough comparison of how different DNN architectures, e.g., CNN, RNN, and Transformer, differ in processing the IoT sensing tasks, which in my opinion, is also an important perspective in such a benchmark.

3. As an important perspective of IoT applications, the benchmark did not mention any efficiency results or considerations.

**Questions:**

1. What is the main difference between the "adapter models" and "unimodal multi-task models"? Do they only differ in the training paradigm, where the adapter models use self-supervised pretraining, while the multi-task models simultaneously optimize for multiple downstream tasks? In my opinion, they are similar because they both utilize a shared encoder to extract the general semantics of a single sensory modality signal.

2. In section 4.2, what are the scales, w.r.t the number of parameters, of each compared model? Do you guarantee that the comparison between different models is fair, by avoiding comparing the performance between models with significantly different scales?

---

### Official Review · Reviewer_S9aC · 2023-11-08

**Soundness:** 1 poor
**Presentation:** 1 poor
**Contribution:** 1 poor
**Rating:** 1
**Confidence:** 4

**Summary:**

This paper claims to present a large multi modality benchmark for Internet-of-things (IoT). There are data present from 12 modalities and 8 tasks are defined to be solved with models trained with this data. The paper further evaluates various different types of architectures to asses how best to combine the various modalities to attain the best accuracy for the various tasks. The motivation for proposing this dataset is because of the claimed need to address various challenges with multimodal data IoT data including "High-modality multimodal learning", "Temporal interactions","Heterogeneity" and "Real-time". Overall the authors find that multi-modality multi-task networks result in the best accuracy on the tasks.

**Strengths:**

Multimodal IoT seems like a potentially interesting under-explored topic.

**Weaknesses:**

The paper is significantly below the acceptance level of ICLR for the following reasons.

1. The paper is poorly written and lacks a clear structure, premise or narrative.
2. It is unclear what the claimed contribution of the work is and how it advances scientific research. It reads more like an opinion piece on the topic of multimodal IoT, rather than offering any concrete scientific insights.
3. Many of the datasets in the collection of 1.115M samples presented in this work are publicly available datasets from other research projects and not ones curated by the authors.
4. The experiments are poorly described and simply not reproducible.
5. The experiments have no clear conclusion or insights.

**Questions:**

I would strongly recommend that the authors review existing published works in ICLR and other AI and computer vision conferences to understand how to improve their papers' presentation, experiments, contributions and style, etc. In its current form the paper is not acceptable as a scientific article.

---

### Meta-Review · Area_Chair_QEeU · 2023-12-05

**Metareview:**

The paper presents a benchmark for multimodality in the context of IoT, with 12 modalities and 8 tasks. While the reviewers found some merit in the dataset, the main finding of the paper that multi-modal multi-task learning provides some gains on this dataset was considered to fall short of being publishable, as there were multiple issues/questions with the framing, presentation of the experiments, and explanation of the experiments. Unfortunately, the authors did not participate in the author/reviewer discussion, and as such the issues have not been clarified. While the paper is recommended to be rejected at this time, the authors are encouraged to take the reviewers' feedback into account for a future submission to make their dataset a valuable contribution to the IoT domain.

**Justification For Why Not Higher Score:**

The paper falls short on multiple fronts such as Writing, Experiments, Contributions, and is far from the acceptance bar.

**Justification For Why Not Lower Score:**

The dataset seems interesting on its own right if it is fleshed out more in the future.

---

### Decision · Program_Chairs · 2024-01-16

Reject